# Evaluation of *CAT* Variants A-89T, C389T, and C419T in Patients with Vitiligo in the Saudi Population

**DOI:** 10.3390/medicina59040708

**Published:** 2023-04-04

**Authors:** Ghada A. Bin Saif, Amal F. Alshammary, Imran Ali Khan

**Affiliations:** 1Department of Dermatology, College of Medicine, King Saud University, Riyadh 11451, Saudi Arabia; 2Department of Clinical Laboratory Sciences, College of Applied Medical Sciences, King Saud University, Riyadh 11433, Saudi Arabia

**Keywords:** vitiligo, *CAT* gene, A-89T, C389T, C419T

## Abstract

*Background and Objectives*: Vitiligo is a chronic autoimmune and depigmentation disorder in humans that manifests as whitening lesions. Reactive oxygen species (ROS) are involved in cell damage. Catalase (CAT) is a well-known oxidative stress regulator and is primarily responsible for the catalytic decomposition of hydrogen peroxide into water and oxygen. Based on previous case-control and meta-analysis studies, we assessed the prevalence of three single-nucleotide polymorphisms (SNPs) of the *CAT* genes A-89T (rs7943316), C389T (rs769217) and C419T (rs11032709) in participants with vitiligo and healthy controls in the Saudi population. *Materials and Methods*: We recruited 152 participants with vitiligo and 159 healthy controls for A-89T, C389T, and C419T SNP genotyping studies using PCR and RFLP analysis. Additionally, we performed linkage disequilibrium and haplotype analyses between vitiligo cases and controls. *Results*: The rs7943316 and rs11032709 SNPs of the *CAT* genes showed a positive association with vitiligo for both heterozygous genotypes and dominant genetic models (TT + AT vs. AA in A-89T and TT + CT vs. CC in C389T), in the *CAT* gene. Linkage disequilibrium analysis revealed a moderate linkage between rs7943316 and rs11032709 SNPs in vitiligo cases and controls. Haplotype frequency estimation revealed a significant association (*p* = 0.003) among the three SNP alleles. *Conclusions*: The rs7943316 and rs11032709 SNPs of the *CAT* genes were strongly associated with susceptibility to vitiligo.

## 1. Introduction

The puzzling disease known as vitiligo (MIM193200) is the most common skin pigmentation disorder characterized by the chronic and progressive loss of melanocytes. The pathophysiology of vitiligo is increasingly being investigated and understood, and novel therapeutic approaches are being developed. The global prevalence of vitiligo is 0.5–2.0% [1,2]. Vitiligo can be acquired or hereditary, and can develop at any age. The clinical features of vitiligo are symmetrical, sometimes single-sided, and often dermatomal milky white spots of varying sizes with variable localization [3]. Age, sex, and ethnicity are not associated with vitiligo; however, some dietary deficiencies are associated with risk factors for the disease [4]. However, the exact cause of the vitiligo remains unclear. Autoimmunity, genetics, keratinocyte death, reduced melanocyte proliferation, neurological dysfunction, and oxidative stress are the hypothesized explanations for its development [5].

Vitiligo, characterized by white macules and patches on the skin and hair, may be segmental, affecting only a small part of the body, or more widespread, affecting all parts of the body. Its stability depends on whether new discolorations appear [6]. In terms of prevalence, vitiligo is not discriminated based on race or sex, making it the leading cause of depigmentation worldwide [7]. There is growing evidence that fibroblasts play a role in inflammatory skin disorders, as they are a common dermal cell type. Although their role in controlling epidermal pigmentation is well-known, their involvement in the communication between immune cells and stromal signals has only been vaguely described [8]. Most patients with vitiligo report feelings of depression and isolation. Several factors have been implicated in the development of vitiligo, including intrinsic and extrinsic melanocyte shortage, oxidative stress, innate immunological inflammation, T-cell-mediated melanocyte destruction, and loss of melanocyte adhesion [9]. However, the mechanisms underlying vitiligo remain largely unclear.

Vitiligo affects people of any age, but usually appears before the age of 20 [10,11]. The consequences of vitiligo on the quality of life of the sufferer are severe. Low self-esteem is a common symptom of vitiligo that has a dominant effect on patients’ social lives and can lead to depression [11]. Differentiating vitiligo subtypes helps to determine the best course of treatment [12]. Vitiligo may be caused by a combination of variables, including genetics, nervous system dysregulation, oxidative stress, biochemicals, minerals, and the immune system [13]. The association between vitiligo and other autoimmune conditions, such as rheumatoid arthritis, multiple sclerosis, primary Sjögren’s syndrome, systemic lupus erythematosus, and inflammatory bowel disease, suggests its autoimmune background. The development of vitiligo involves both cellular and humoral immune responses. There is also evidence that melanocytes have difficulty adjusting to stress, which may lead to their death due to the triggering of the immune system [14]. Several loci encoding non-major histocompatibility complex-regulating immunoproteins have been shown to be associated with generalized vitiligo, suggesting that vitiligo may have a genetic basis [15].

Several factors have been estimated to contribute to the development of vitiligo. Approximately 23% of monozygotic twins share a common case of widespread vitiligo. Relatives of people with vitiligo have a 6–18-fold increased risk of developing this condition. In addition, an epidemiological study linked the development of vitiligo to single-nucleotide polymorphisms (SNPs) in several genes associated with this condition. Therefore, genetic predisposition seems to be an important endogenous element in the development of vitiligo [16]. Catalase (CAT; EC 1.11.1.6) is an enzyme that catalyzes the breakdown of hydrogen peroxide (H_2_O_2_) into water and oxygen, neutralizing the extremely reactive free radicals produced in the process and protecting cells from damage. Patients with vitiligo show decreased catalase activity and increased deposition of excess H_2_O_2_ throughout the epidermis; hence, *CAT* gene mutations have been implicated in vitiligo [17]. More than 40 susceptibility loci, including *MC1R*, *TYR*, *IFIH1*, *CD44*, *CD80*, *GZMB*, *HLA-A*, *XBP1*, *CAT*, and *MTHFR*, have been elucidated to be linked to vitiligo through large genome-wide association studies in the past decade [18,19]. The loss of melanocytes involves the interaction of several genes and immunological and environmental events. Of these, only a limited number of candidate genes have been associated with autoimmunity, while others may be connected to environmental events such as oxidative stress. *CAT* is located on chromosome 11p13 and consists of 13 exons and 12 introns [20]. Several mutations in *CAT* have been associated with vitiligo symptoms [21]. The SNP A-89T (rs7943316) in the promoter region of *CAT* can influence transcription rates, resulting in lower CAT expression. C389T (rs769217) is present in exon 9, while C489T (rs11032709) is present in exon 10 and is also a silent substitution that has been found to be associated with vitiligo [22]. Few studies have investigated the association between vitiligo susceptibility and SNPs in the *CAT* gene, and in the Saudi population, it has not been investigated so far.

Therefore, the current study aimed to evaluate the prevalence of SNPs in the *CAT* gene and vitiligo in Saudi patients.

## 2. Materials and Methods

### 2.1. Ethical Issues

Ethical issues were resolved by obtaining an ethics grant from the Institutional Review Board of King Saud University (E-20-4773; 04/10/2020). A total of 311 patients signed informed consent forms prior to participating in the study. This study was carried out as per the Declaration of Helsinki.

### 2.2. Study Participants 

This study was designed after calculating the minimum sample size. It included 311 participants, of whom 152 had vitiligo and 159 were healthy controls. This study was conducted throughout 2015 (12 months). The participants were unrelated in terms of age or sex. This study was retrospective, and 152 vitiligo patients and 159 healthy controls were selected based on our previous study [23]. Vitiligo patients were included if diagnosed by two dermatologists based on the diagnostic criteria of the Vitiligo European Task Force with a 100% diagnostic agreement [23]. Pregnant women and participants with any unclassified vitiligo, those under any medication, or those affected by other autoimmune diseases were excluded from this study [23]. Healthy controls without any other diseases, specifically autoimmune diseases, were included in this study. The exclusion criteria for the controls were a diagnosis of any human disease or a family history of vitiligo. Participants willing to provide only oral consent were excluded. Only participants (both cases and controls) that provided written informed consent were included in the study.

Age, sex, and family history data from all 311 participants were recorded [23], and 2 mL of EDTA blood was collected from each participant.

### 2.3. Analysis of Nucleic Acid

Genomic DNA from the 311 EDTA peripheral blood samples was extracted using a Qiagen DNA isolation kit according to the manufacturer’s protocol (Cat#51104, Lot#157037757; 40,724 Hilden, Germany), and the extracted genomic DNA was measured using a NanoDrop spectrophotometer for assessing DNA quality (Thermoscientific, NANODROP ONE^C^, S. No-AZY1707157; Thermo Fischer Scientific, Madison, WI, USA). DNA was stored at −80°C until polymerase chain reaction (PCR) for the genotyping of the *CAT* gene. SNPs in the *CAT* gene are associated with oxidative stress and reactive oxygen species (ROS) scavenging, and based on previous studies and bioinformatic analyses, the prevalence of three SNPs, namely, A89T (rs7943316), C389T (rs769217), and C419T (rs11032709), was investigated in patients with vitiligo. The complete details of the SNPs, primers, and precise restriction enzymes are shown in Table 1.

Genotyping of the three SNPs was performed using PCR (Applied Biosystems, Model # 9902, Serial # 2990210130, Singapore), and the purified products were digested with suitable restriction enzymes (New England Biolabs, Ipswich, UK). PCR was conducted using 10 ng of genomic DNA with 10 pmol of both forward and reverse primers, Qiagen master mix (Qiagen Taq PCR master mix; CAT# 201445; 40,724 Hilden, Germany), and double-distilled water for a final volume of 50 µL in 0.2 mL PCR tubes. The cycling conditions were as follows: initial denaturation at 95 °C for 5 min, followed by denaturation at 95 °C for 30 s, annealing at 60 °C for A-89T, 62 °C for C389T, 64 °C for C419T, and extension at 72 °C for 30 s, followed by a final extension at 72 °C for 5 min. PCR (35 cycles) was performed using an Applied Biosystems thermal cycler (Applied Biosystems, Model # 9902, Serial # 2990210130, Singapore) for approximately 86 min. Primers were purchased from Macrogen (Macrogen, Inc., Korea). The PCR products were analyzed on 2% agarose gel stained with ethidium bromide (Lonza, SeaKem^®^ LE Agarose, CAT # 50004, Rockland, NY, USA). PCR products were then digested with specific restriction enzymes (from New England Biolabs, UK) at 37 °C for 16 h and separated on a 3% ethidium bromide-stained agarose gel. The PCR and RFLP bands for all three SNPs are shown in Figure 1.

### 2.4. Statistical Analysis

Continuous data were represented as the mean ± standard deviation (m ± SD), and discrete data were expressed as percentages (Table 1). The chi-square test was used to assess deviation from Hardy–Weinberg equilibrium (HWE) in the controls for the three SNPs. HWE analysis was performed using Microsoft Excel [24]. SNPstat was used to calculate the genotypes, alleles, and various modes of inheritance using odds ratios, 95% confidence intervals (CIs), and haplotype analysis between the vitiligo cases and the controls [23]. The pairwise linkage disequilibrium (LD) coefficient (D’) between the three SNPs was calculated using HaploView (version 4.2). *p* ≤ 0.05 was considered statistically significant.

## 3. Results

### 3.1. Characteristics of Vitiligo Patients

A total of 152 patients with generalized vitiligo (26.01 ± 13.2) and 159 older healthy controls (43.6 ± 17.8) were recruited. The vitiligo cases and healthy controls showed non-significant associations (*p* = 0.62). The male and female participants were 42.1% and 57.9%, respectively. Approximately 31% of patients with vitiligo had documented self- and family histories, whereas 20.4% were confirmed for consanguineous parents. Table 2 describes the ages, sexes, and family histories of the patients with vitiligo.

### 3.2. HWE and Genotyping Analysis for CAT Gene SNPs

Three SNPs were successfully selected and genotyped in the cases and controls, with a success rate of over 99%. Table 3 shows the allelic genotype distributions of vitiligo cases and controls. Genotypic frequencies for the A-89T, C389T, and C419T polymorphisms in the two groups were found to be consistent with HWE analysis—A-89T in vitiligo cases (X^2^ = 23.32; *p* = 0.0001) and controls (X^2^ = 1.20; *p* = 0.2701), C389T in vitiligo cases (X^2^ = 23.42; *p* = 0.0001) and controls (X^2^ = 5.66; *p* = 0.99), and C419T in vitiligo cases (X^2^ = 0.001; *p* = 0.9612) and controls (X^2^ = 0.16; *p* = 0.6804).

The genotypic frequencies among the A-89T polymorphisms as AA, AT, and TT genotypes were 11.2%, 69.1%, and 19.7% in vitiligo cases and 19.5%, 54.1%, and 26.4% in controls, respectively. A total of 19.7% of vitiligo patients had TT genotypes, which was a lower proportion than in the controls (26.4%). Homozygous TT vs. AA (X^2^ = 3.70, OR-1.31; 95%CI:0.61–2.77; *p* = 0.4920) and heterozygous (X^2^ = 5.87,OR-2.26; 95%CI:1.15–4.29; *p* = 0.0153) and dominant models (AA vs. AT + TT; X^2^ = 4.11, OR-1.92; 95%CI:1.01–3.64; *p* = 0.0425) showed a positive association, while homozygous TT (X^2^ = 3.70;OR-1.31; 95%CI:0.61–2.77; *p* = 0.4920) and other genetic models (AA + AT vs. TT; OR-0.52; 95%CI:0.33–0.83; *p* = 0.006 and AA + AT vs. TT; X^2^ = 7.37, OR-0.52; 95%CI:0.33–0.83; *p* = 0.0066) showed a negative association. In both the vitiligo cases and the controls, similar allelic frequencies (T = 0.54 and A = 0.46) indicated non-significant association (T vs. A; X^2^ = 0.04,OR-1.03; 95%CI:0.75–1.41; *p* = 0.8381).

In the C389T polymorphism, the CC and TT genotypic frequencies were significantly higher in the controls (21.4% and 28.9%, respectively) than in the vitiligo cases (9.2% and 22.4%, respectively), whereas the CT genotypic frequency was higher in the vitiligo cases (68.4%) than in the controls (49.7%). The risk and protective alleles were 0.57 and 0.43, respectively, in vitiligo cases, and 0.54 and 0.46, respectively, in controls. When vitiligo cases and controls were assessed, there was a non-significant association with allele frequency (T vs. C; X^2^ =0.49, OR-1.12; 95%CI:0.81–1.53; *p* = 0.4819). Only the dominant model showed a positive association between the vitiligo cases and the control subjects (CC vs. CT + TT; X^2^ = 8.82, OR 8.82; 95%CI: 1.37–5.22; *p* = 0.0029).

Yates’ correction was applied to the C419T polymorphism because of the absence of TT genotypes in both the vitiligo cases and controls. CT genotypes were higher in the controls (6.3%) than in the vitiligo cases (0.7%). CC genotypes were found in 99.3% of patients with vitiligo and 93.7% of the controls. The results of the association analysis revealed that C419T polymorphism was a protective factor against vitiligo. When vitiligo cases were compared with healthy controls, the risk of vitiligo decreased. None of the genotypes (CT vs. CC; X^2^ = 7.22, OR-0.09; 95%CI:0.01–0.78; *p* = 0.0071; TT vs. CC; X^2^ = 7.16, OR-0.98; 95%CI:0.01–50.04; *p* = 0.9947 *), genetic models (CC vs. CT + TT and CC + TT vs. CT; X^2^ = 7.22, OR-0.09; 95%CI:0.01–0.78; *p* = 0.0071 (*) and CC + CT vs. TT; OR- X^2^ = 0.95, 1.05; 95%CI:0.02–53.38; *p* = 0.9821 *), or allele frequencies (T vs. C; X^2^ = 7.09, OR-0.10; 95%CI:0.01–0.79; *p* = 0.0077) were positively associated with the vitiligo cases nor with the controls.

### 3.3. Linkage Disequilibrium Analysis

Table 4 and Table 5 describe the LD analysis for both the vitiligo cases and controls. The delta coefficient (D’) was calculated for both vitiligo patients and controls for three SNPs, namely, rs7943316, rs769217, and rs11032709, in the *CAT* gene. The results confirmed a significant association (*p* < 0.0001) between the three alleles segregated together, as they played a strong role in patients with vitiligo, as well as among the controls from the Saudi population.

LD analysis revealed a moderate linkage between rs7943316 and rs11032709 in the controls, but a moderate linkage between rs7943316 and rs769217 in vitiligo cases. Moderate linkage disequilibrium was observed between rs7943316, rs769217, and rs11032709 SNPs (Figure 2).

### 3.4. Haplotype Analysis

In Table 6, The analysis of the haplotype frequency estimation revealed a significant association (*p* = 0.003). However, there was a negative association with haplotype association in response to any of the alleles in Table 7.

## 4. Discussion

Based on previous studies, we chose three SNPs (A-89T (rs7943316), C389T (rs769217), and C419T (rs11032709)) to investigate their associated risk of vitiligo in the Saudi population [22]. Moreover, the A89T and C319T SNPs were associated. Linkage disequilibrium analysis revealed a moderate linkage between the rs7943316 and rs11032709 SNPs in both the vitiligo cases and the controls (Figure 2). The analysis of haplotype frequency estimation revealed the significant association (*p* = 0.003; Table 6) and no association between the haplotype and response to the other alleles (Table 7).

The SNPs A-89T (rs7943316), C389T (rs769217), and C489T (rs11032709) in the global population with vitiligo were found to have both positive and negative associations [20,21,22,25,26,27,28,29,30,31,32,33,34,35,36]. The heterozygous genotype was substantially related to the A-89T (rs7943316) and C389T (rs769217) SNPs in our study, and similar results have been reported previously [22,25,29,33]. In our study, among A-89T (rs7943316) and C389T (rs769217), the TT genotype was found to be more prevalent in controls and to confer protection against vitiligo [21,27,31,34]. The C419T (rs11032709) SNP was not associated with vitiligo in the Saudi population, which is in line with the results of previous studies [22,25,27,29,34]. In this study, an investigation of the A-89T (rs7943316) SNP revealed that the AT genotype was associated with an increase in the risk of developing vitiligo by 2.26 times in the Saudi population. This risk was documented to be 1.92 and 1.03 times higher in the dominant model and allele frequencies, respectively. The C389T (rs769217) SNP with the CT genotype was associated with an increase in the risk of developing vitiligo by 3.19 times, and by 2.68 times in the dominant model.

A slower proliferation of melanocytes was observed in patients with vitiligo than in controls, and a dysregulated redox balance was associated with decreased expression of catalase. In particular, melanocytes generate substantial amounts of ROS as a byproduct of melanin formation, and catalase shields cells from this damage by breaking down hydrogen peroxide into oxygen and water (H_2_O_2_ = O_2_ and H_2_O). Hence, growth factors and catalase, as well as other compensating media, are needed to culture melanocytes from patients with vitiligo [37]. Because vitiligo patients have been found to have decreased catalase enzyme activity throughout their epidermis, *CAT* is a candidate gene in vitiligo, and was selected for our study. Many allelic variants of *CAT* have been documented, with the first described form of acatalasemia tracing back to a splicing mutation prevalent in the Japanese population. Whole cDNA sequencing of *CAT* revealed a 1581 bp coding region. While vitiligo-like depigmentation has not been reported as a phenotypic feature of acatalasemia or hypocatalasemia, reports of low epidermal catalase activity in both lesioned and non-lesioned skin of patients with vitiligo with concomitant increases in epidermal H_2_O_2_ levels have suggested a possible role of catalase in vitiligo susceptibility. Catalase deficiency in the skin of patients with vitiligo may be caused by tissue-specific changes in gene expression or enzyme structure/function in melanocytes and/or keratinocytes. Catalase mRNA levels in cultured melanocytes from the lesional or non-lesional epidermis showed no significant changes in either the patients with vitiligo or the healthy controls. Preliminary data from case–control and family studies have revealed a genetic link between *CAT* and vitiligo susceptibility, suggesting that reduced catalase enzyme activity is caused by mutations in the *CAT* gene [25].

A meta-analysis [30] showed that the C389T (rs769217) SNP was not associated with vitiligo in Caucasians or in Asians. A case–control study demonstrated a negative association [30]. Another meta-analysis [20] confirmed a negative association between C389T (rs769217) in *CAT* and vitiligo. However, in the European population, the CT genotype might be a risk factor, whereas the CC genotype may act as a protective factor, for vitiligo. Lv et al. conducted the first meta-analysis analyzing four case–control studies of vitiligo cases with the C389T SNP, and discovered a positive association [32]. Mehaney et al. found a negative association between the C389T (rs769217) SNP and the *CAT* gene in Egyptian vitiligo patients [21]. A study on a Korean population found a negative association using the restriction enzymes *Hinf*I and *BstX*I, as well as a risk association using haplotype analysis [35].

One of the limitations of our study was the lack of serum analysis. In our study, the controls were older than the vitiligo patients. We excluded the fourth SNP C-262T in *CAT,* which has also been linked to vitiligo, due to lower band sizes after digestion with *Sma*I between the C (143/38 + 9 bp) and T (143/47 bp) genotypes.

## 5. Conclusions

In conclusion, this study confirmed the significant association between heterozygous genotypes and dominant models of A-89T (rs7943316) and C389T (rs769217) SNPs in the *CAT* gene. C419T (rs11032709) SNP was not found to be associated with vitiligo in the Saudi population. However, additional statistical analysis should be performed in studies with large sample sizes, and meta-analysis studies are recommended for the study of the remaining SNPs and their relationship with vitiligo in the *CAT* gene.

## Figures and Tables

**Figure 1 medicina-59-00708-f001:**
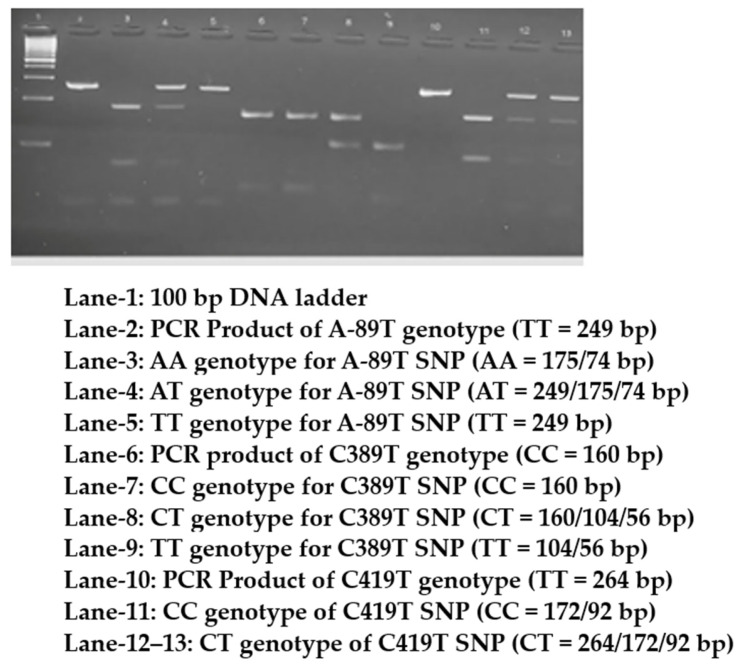
Digested and undigested PCR products of *CAT* gene genotypes separated on a 2.5% agarose.

**Figure 2 medicina-59-00708-f002:**
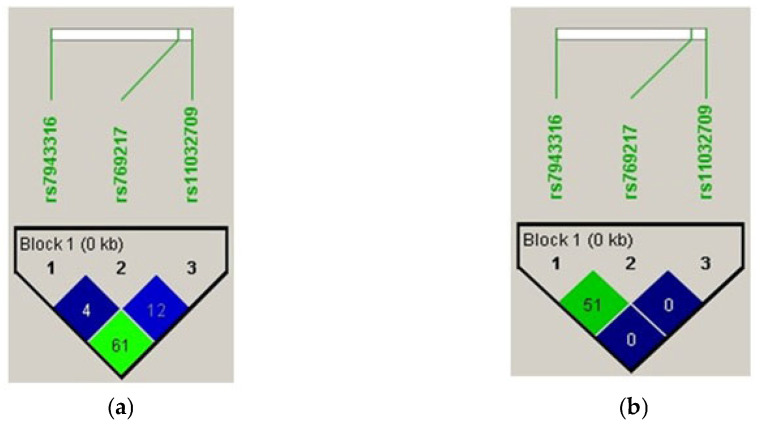
Pairwise linkage disequilibrium among the three SNPs in (**a**) vitiligo cases and (**b**) controls.

**Table 1 medicina-59-00708-t001:** List of SNPs, primers, and restriction enzymes involved in this study.

SNPs	rsID Number	Forward Primer	Reverse Primer	PCR	Restriction Enzyme	Band Sizes
A-89T	rs7943316	5′-AATCAGAAGGCAGTCCTCCC-3′	3′-TCGGGGAGCACAGAGTGTAC-5′	249 bp	*Hinf*I	A-175/74 bp; T-249 bp
C389T	rs769217	5′-CCTTTTTGCCTATCCTGACACTCAC-3′	3′-AGGGGGAGCCCAACGTCTTTAG-5′	190 bp	*BSTX*1	C-160 bp; T-104/56 bp
C419T	rs11032709	5′-CCTAAGTGCATCTGGGTGGT-3′	3′-TACATCAGACAGTTGGGGCA-5′	264 bp	*BstN*1	C-172/92 bp; T-264 bp

**Table 2 medicina-59-00708-t002:** Comparison of clinical characteristics between vitiligo cases and controls.

	Vitiligo Patients (*n* = 152)	Control (*n* = 159)	*p*-Value
Gender			
Male	64 (42.1)	67 (42.1)	0.80
Female	88 (57.9)	92 (57.9)	0.77
Age (years)	26.01 ± 13.2	43.6 ± 17.8	0.62
Disease duration (years)	4.1 ±2.6	NA	NA
Generalized clinical type of disease	152 (100%)	NA	NA
Family history	47 (30.9%)	NA	NA
Consanguinity	31 (20.4%)	NA	NA

NA = Not applicable/no analysis.

**Table 3 medicina-59-00708-t003:** Comparison of genotypic and allele frequencies between vitiligo cases and controls.

	Vitiligo (*n* = 152)	Controls (*n* = 159)	X^2^ Value	OR	95% CI	*p*-Value
A-89T (rs7943316)						
AA	17 (11.2%)	31 (19.5%)	Reference	Reference	Reference	Reference
AT	105 (69.1%)	86 (54.1%)	5.87	2.26	1.15–4.29	0.0153
TT	30 (19.7%)	42 (26.4%)	3.70	1.31	0.61–2.77	0.4920
AA vs. AT + TT	135 (88.8%)	128 (80.5%)	4.11	1.92	1.01–3.64	0.0425
AA + TT vs. AT	47 (30.9%)	73 (45.9%)	7.37	0.52	0.33–0.83	0.0066
AA + AT vs. TT	122 (80.3%)	117 (73.6%)	1.94	1.46	0.85–2.48	0.1629
A	139 (0.46%)	148 (0.46%)	Reference	Reference	Reference	Reference
T	165 (0.54%)	170 (0.54%)	0.04	1.03	0.75–1.41	0.8381
HWE	0.54	0.53				
X^2^	23.32	1.20				
*p*-value	0.0001	0.2701				
C389T (rs769217)						
CC	14 (9.2%)	34 (21.4%)	Reference	Reference	Reference	Reference
CT	104 (68.4%)	79 (49.7%)	11.65	3.19	1.60–6.34	0.0006
TT	34 (22.4%)	46 (28.9%)	4.58	1.79	0.83–3.85	0.1315
CC vs. CT + TT	138 (90.8%)	125 (78.6%)	8.82	2.68	1.37–5.22	0.0029
CC + TT vs. CT	48 (31.6%)	80 (50.3%)	11.26	0.45	0.28–0.72	0.0007
CC + CT vs. TT	118 (78.6%)	113 (71.1%)	1.75	1.41	0.84–2.35	0.1859
C	132 (0.43%)	147 (0.46%)	Reference	Reference	Reference	Reference
T	172 (0.57%)	171 (0.54%)	0.49	1.12	0.81–1.53	0.4819
HWE	0.57	0.54				
X^2^	23.42	5.66				
*p*-value	0.0001	0.99				
C419T(rs11032709)						
CC	151 (99.3%)	149 (93.7%)	Reference	Reference	Reference	Reference
CT	01 (0.7%)	10 (6.3%)	7.22	0.09	0.01–0.78	0.0071
TT	00 (0%)	00 (0%)	7.16^*^	0.98	0.01–50.04	0.9947 *
CC vs. CT + TT	01 (0.7%)	10 (6.3%)	7.22	0.09	0.01–0.78	0.0071
CC + TT vs. CT	151 (99.3%)	149 (93.7%)	5.66 *	0.09	0.01–0.78	0.0071 *
TT vs. CC + CT	152 (100%)	159 (100%)	0.95	1.05	0.02–53.38	0.9821 *
C	303 (99.7%)	308 (96.8%)	Reference	Reference	Reference	Reference
T	01 (0.3%)	10 (3.2%)	7.09	0.10	0.01–0.79	0.0077
HWE	0.00	0.03				
X^2^	0.001	0.16				
*p*-value	0.9612	0.6804				

* indicates Yates correction.

**Table 4 medicina-59-00708-t004:** Analysis of linkage disequilibrium between vitiligo patients and the three SNPs.

L1	L2	D’	LOD	r^2^
rs7943316	rs769217	0.518	2.17	0.202
rs7943316	rs11032709	0.0	0.0	0.0
rs769217	rs11032709	0.0	0.0	0.0

0.0 indicates < 0.0001.

**Table 5 medicina-59-00708-t005:** Analysis of linkage disequilibrium between the three SNPs and controls.

L1	L2	D’	LOD	r^2^
rs7943316	rs769217	0.044	0.03	0.001
rs7943316	rs11032709	0.619	0.45	0.014
rs769217	rs11032709	0.123	0.01	0.0

0.0 indicates < 0.0001.

**Table 6 medicina-59-00708-t006:** Haplotype analysis of frequency estimation.

S. No	A-89T	C389T	C419T	Total	Group 0	Group 1	Cumulative Frequency
1	T	T	C	0.3016	0.2714	0.38	0.3016
2	T	A	C	0.2467	0.2605	0.1858	0.5478
3	C	T	C	0.2226	0.2376	0.1628	0.7708
4	C	A	C	0.2115	0.1991	0.2681	0.9823
5	C	T	T	0.0145	0.0256	0	0.9968
6	T	A	T	0.0032	0.0059	NA	1
7	C	A	T	0	0	0.0033	1

**Table 7 medicina-59-00708-t007:** Haplotype association with response (*n* = 311, crude analysis).

S. No	A-89T	C389T	C419T	Frequency	OR (95% CI)	*p*-Value
1	T	T	C	0.3108	1.00	-
2	T	A	C	0.2377	0.79 (0.42–1.48)	0.46
3	C	A	C	0.2207	0.90 (0.54–1.50)	0.68
4	C	T	C	0.2131	0.74 (0.39–1.39)	0.34
5	C	T	T	0.0147	0.11 (0.01–0.90)	0.041
rare	*	*	*	0.0029	0.00 (Inf-Inf)	1

* indicates rare; Global haplotype association (*p* = 0.095).

## Data Availability

All of the obtained data are present in the manuscript. If an additional analysis datasheet is required, the corresponding author provide one upon reasonable request.

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
