# Peer review of "Evaluation of CAT Variants A-89T, C389T, and C419T in Patients with Vitiligo in the Saudi Population"

_medicina, 2023, doi:10.3390/medicina59040708_

Round 1
Reviewer 1 Report
Title must not have apparent spelling mistakes
Introduction: English needs to be scientific. Few sentence doesn’t carry any meaning as marked.
Enough details not given as why they have selected this three SNPs for the respective study. Whereas,
at the end of the discussion, fourth SNP has been mentioned which has not been mentioned above.
2. Materials and Methods: The reference mentioned as “our recent publication” in Line 117 for
collection of vitiligo samples does not carry any sample collection data.
2.3 Analysis of nucleic acid, Line 135, author is not sure why they have selected the SNP. And, no
reference of the bioinformatics analysis performed
2.4: Validation: Not clear.
3. Results: Line 175: Anthropometric measurement doesn’t include male,female,age.
Line 192: Vitiligo patients showed lower level of TT and AA compare to Healthy Controls.
Line 196: For recessive model, you have put the heterozygous value. Interpretation needs
modification.
Line 216: “High vitiligo patients” term meaning is not clear.
3.3 Linkage disequilibrium analysis: Mention the cut off for considering the LD value significant or
moderately significant.
4. Discussion:
Line 254: Sentence doesn’t carry any meaning.
Line 265-266: No proper explanation has been provided
Line 274-275: Haplotype frequency of which allele has not been explained.
References after 27 are missing in the manuscript. Thus data could not be matched or checked.
Overall comment:
The writing needs to be improved and scientific. The study you conducted to examine the association
between variants in the CAT gene and vitiligo among Saudi Arabs is a good initiative. There is a need
to explain why you chose only CAT and not SOD or other oxidative stress genes, and why only these
three SNPs. You must explain how the genotype and/or allele that you obtained to be significant
contribute to the disease vitiligo in the Discussion.
Author Response
Reviewer Comments 1
Title must not have apparent spelling mistakes
Title was updated as per your recommendation
Introduction: English needs to be scientific. Few sentences doesn’t carry any meaning as marked.
Manuscript was edited with native experts
Enough details not given as why they have selected this three SNPs for the respective study. Whereas, at the end of the discussion, fourth SNP has been mentioned which has not been mentioned above.
Previous studies including meta-analysis studies has confirmed Vitiligo patients may have a problem manufacturing catalase in their body or delivering catalase to the skin. However, there was no documented studies in the Saudi Population and then we have designed this study
- Materials and Methods: The reference mentioned as “our recent publication” in Line 117 for collection of vitiligo samples does not carry any sample collection data.
Our intention was to mention as “Based on previous publication” same number of sample size was included for this study also. We have updated the sentence and details were given
2.3 Analysis of nucleic acid, Line 135, author is not sure why they have selected the SNP. And, no reference of the bioinformatics analysis performed
As we have discussed earlier, there was a relation between CATALASE gene and Vitiligo and based on previous publication, we have selected these SNPs
A-89T- Liu et al (2010). Promoter Variant in the Catalase Gene Is Associated with Vitiligo in Chinese People
C389T- He et al (2015). Lack of association between the 389C>T polymorphism (rs769217) in the catalase (CAT) gene and the risk of vitiligo: An update by meta-analysis
C419T- Liu et al (2010). Promoter Variant in the Catalase Gene Is Associated with Vitiligo in Chinese People.
2.4: Validation: Not clear.
Digested PCR products were run on 2.5% agarose gel and image was captured with mobile.
- Results: Line 175: Anthropometric measurement doesn’t include male, female, age.
Thank you for your comment. We have rectified it.
Line 192: Vitiligo patients showed lower level of TT and AA compare to Healthy Controls.
We have modified the sentence.
Line 196: For recessive model, you have put the heterozygous value. Interpretation needs modification.
We have updated the values
Line 216: “High vitiligo patients” term meaning is not clear.
It was “High in vitiligo patients”. We have updated it.
3.3 Linkage disequilibrium analysis: Mention the cut off for considering the LD value significant or moderately significant.
0.0 indicates <0.0001. It was updated in the tables 4 and 5
- Discussion:
Line 254: Sentence doesn’t carry any meaning.
We have removed the sentence based on your recommendation
Line 265-266: No proper explanation has been provided
Now the sentence was updated to understand properly.
Line 274-275: Haplotype frequency of which allele has not been explained.
The details were given in Table-6
References after 27 are missing in the manuscript. Thus, data could not be matched or checked.
Reference 27 was updated (Spritz et al 2012).
Spritz, R.A. Six decades of vitiligo genetics: genome-wide studies provide insights into autoimmune pathogenesis. Journal of investigative dermatology 2012, 132, 268-273
Overall comment:
The writing needs to be improved and scientific. The study you conducted to examine the association between variants in the CAT gene and vitiligo among Saudi Arabs is a good initiative. There is a need to explain why you chose only CAT and not SOD or other oxidative stress genes, and why only these three SNPs. You must explain how the genotype and/or allele that you obtained to be significant contribute to the disease vitiligo in the Discussion.
Dear Editor,
Thank you for your feedback. We really appreciate your effort. We have divided to screen limited SNPs/variants separately. For example, we have published VDR gene as a single manuscript, now, we have submitted CATALASE gene and in future we have other genes in which SOD is also included. We want to publish separately. These 3 SNPs (A-89T, C389T and C419T) were previously published with vitiligo disease. For your reference, I am citing the references below. We have explained the role of genotype and allele frequencies in the discussion.
References
A-89T- Liu et al (2010). Promoter Variant in the Catalase Gene Is Associated with Vitiligo in Chinese People
C389T- He et al (2015). Lack of association between the 389C>T polymorphism (rs769217) in the catalase (CAT) gene and the risk of vitiligo: An update by meta-analysis
C419T- Liu et al (2010). Promoter Variant in the Catalase Gene Is Associated with Vitiligo in Chinese People
Our Team is thankful to “Prince Naif Health Research Centre, Investigator Support Unit for providing the Language editing services”

Reviewer 2 Report
The article entitled “Evaluation of Catalase genetic variants associated with Vitiligo in Saudi subjects” Presents the result of the association analysis of three SNPs in the Catalase gene and the presence of Vitiligo in this population.
Although the article could be relevant in terms of the genetic component involved in the altered processes in this disease, the authors must attend to some errors in the content and in the interpretation of the results.
About, It is suggested that the SNPs analyzed be included in the title.
In summary:
The writing needs to be improved.
Line 17 Remove (Vit-uh-lie-go)
The objective must be clear… Analyze the SNPs (A-89T, C389T and C419T) of the CAT gene in subjects with vitiligo from the Saudi population. Line 21-22
The results must be clear:
Line 28-29 page 1:
Haplotype frequency estimation revealed a significant association (p=0.003) and no association with haplotype association with response with any of the alleles. It is not clear what the authors are trying to say.
Line 30-32 In the conclusion, the description of the polymorphisms must be clear… A-89T, C389T It is recommended to use the “rs” to identify each polymorphism.
Introduction
Page 1 line 44… Vitiligo is categorized into segmental and non-segmental vitiligo. REPLACE BY non-segmental forms.
Page 1 line 46: (Bin-Saif et 46 to 2021) put correct citation format in the text.
Page 2 line 55 says CATALASE and should say Catalase. Remember to correctly place protein and gene names throughout the text.
Page 2 lines 68 and 69 include the rs of the 3 SNPs to analyze.
Matodologia: The writing must be improved
Page 2 line 75-76: The participants who were willing to give the oral consent form were excluded from this study. This line can be removed or is only included as exclusion criteria.
The participant section must be clearly written. The part of signing the informed consent and ethics committee approval can be included at the end of this section.
Page 2 line 83 pass (Bin-Saif et al 2021) to correct reference format.
Add after this clearly the inclusion criteria.
Page 2 line 90 and 91: In this study, 2 ml of EDTA blood was collected after the signature obtained from the patient on the informed consent form. Remove, and only leave the content in nucleic acid analysis.
Page 2 line 95 and 96 the citation of the real Qiagen protocol should be added and the citation of (Bin-Saif et al 2021) removed.
Page 2 line 98 to 100, page 3 line 101
The SNPs in the CAT gene are responsible for several oxidative stress and reactive oxygen species and based on previous studies or bioinformatic analysis, three SNPs such as A89T, 389T and C419T were investigated in vitiligo subjects.
It does not provide additional information. Just indicate that the 3 SNPS are going to be analyzed and add their respective “Rs”.
Genetic analysis:
Page 3 lines 102 to 103:
The PCR protocol must be clear, it must indicate the number of PCR cycles, alignment time of the primers, describe the origin of all the reagents (Primers, restriction enzymes), the brand and model of the equipment used (Thermocycler, Photo-Doc, etc).
You must replace “run” with “analyzed”
Page 3 lines 115 to 119. In the validation you must indicate the type of sequencing equipment used, and preferably include an example image of some electropherogram obtained with the polymorphism analyzed. Furthermore, the reference must correspond exactly to the method; you point out that it is Based on Bogari et al (reference 9), but in that article other authors are cited: “Validation was performed with Sanger sequencing analysis (Fig. 2) based on our recent publication (Alharbi et al., 2021)”.
Page 3 lines 120 to 130. Statistical analysis. The writing needs to be improved. The statistical tests used must be clearly explained: for continuous and discrete variables ( continuous variables, indicate whether normality tests were performed before deciding to use the Student's T test). The reference used to support the HW equilibrium calculation is spurious.
Include support reference for SNPstat, and I suggest reviewing a similar article on the subject.
Results
The interpretation of the results should be improved.
Page 3 lines 132 to 139 and table 2. The interpretation of the lack of difference in the ages of the vitiligo and controls is not an association result, it should only be used to support that the following analyzes are correct, since there are no differences in the study group.
Table3. It is necessary to review the interpretation of the P-value in HW equilibrium, since the P-values for two polymorphisms in Vitiligo cases are not in equilibrium.
On the other hand, for the polymorphism analysis it would be important to consider including the chi2 values in the comparison made for 2 groups in each of the models and in the comparison of the alleles.
In addition, it is recommended to review the approximation of the values; For example, the authors report for rs7943316 AA vs AT a value of p= 0.01, and it really is p=0.01539, that is, it is close to p=0.02.
In alleles of the first 2 polymorphisms, the % symbol is missing.
Page 5, lines 153 to 186 it is not necessary to repeat the results of the table. It is recommended to limit to describing only relevant data. In this regard, the description is confusing and has errors in interpretation. For example:
Do not repeat genotype and allele frequency results for each polymorphism.
Lines 158 to 160 says:
and homozygous TT (OR-1.31; 95%CI: 0.61-2.77; p=0.49) and other genetic models (AA+AT vs TT; OR-0.52; 95%CI: 0.33-0.83; p=0.006 and AA+AT vs TT; OR-1.46; 95% CI: 159 0.85-2.48; p=0.16) showed the negative association.
And should say for the genetic model:
“models (AA+TT vs AT;OR-0.52; 95%CI: 0.33-0.83; p=0.006 and AA+AT vs TT OR-1.46; 95%CI: 159 0.85-2.48; p=0.16), where in the first if there is association”.
Page 5 line 170 says p=01.8 correct
Line 17º1 to 175 it is not necessary to describe information that is already in the table. On the other hand, the authors should explain their interpretation of risk and protective allele.
For rs11032709 there are only 2 genotypes. We suggest removing the TT results and the analyzes of genetic models that include them from Table 3. They do not provide information and cannot suggest models. For this reason, lines 176 to 186 must be corrected and redrafted.
In Linkage Disequilibrium and Haplotype analysis it is suggested to explain what they mean by their results, if this affects their analysis. Once again, the text is only a repetition of the result present in tables 4 to 7, and does not provide additional information to understand the objective of its analysis.
Figure 1. The additional information for each of the samples must be in text format as a caption, not as part of the photograph. It is recommended to describe characteristics of the molecular weight marker, and its representative sizes must be described in the image.
For Figure 2, it must contain a sufficiently descriptive image caption, in addition, the image format, separated for one analysis and the other, does not provide relevant information. They should indicate Figures 2a and 2b, each with an explanation of its content.
Discussion:
The discussion should contain the contrast of the results obtained in this work with what is already published.
In this regard lines 200 to 215 can be deleted. Lines 215 to 230 is a repetition of the results.
Only from line 231 is there a real discussion of the work.
Line 238 still needs to be added reference to He et al meta-analysis studies.
Line 251 to 253 actually describe the result of the study carried out in the Korean population and how it compares with what was obtained in this investigation.
Lines 254 to 259 delete. Just leave a section of study limitations.
The conclusion should be improved
If the authors consider the reference genotypes, an association is observed in each polymorphism. On the other hand, they could mention that considering certain genetic models, there is also an association between genotypes and the presence of vitiligo.
Lines 268 to 270 should be removed as it is not a conclusion to your work.
The bibliography should be reviewed. There are lost references in the text, and others that do not correspond (example reference 10)
Author Response
Reviewer 2
The article entitled “Evaluation of Catalase genetic variants associated with Vitiligo in Saudi subjects” Presents the result of the association analysis of three SNPs in the Catalase gene and the presence of Vitiligo in this population. Although the article could be relevant in terms of the genetic component involved in the altered processes in this disease, the authors must attend to some errors in the content and in the interpretation of the results.
Dear Reviewer, Thank you for your feedback. We will incorporate all your valuable comments in the revised manuscript. After modifying the title, page number was shifted.
About, it is suggested that the SNPs analyzed be included in the title.
We have updated the title
In summary: The writing needs to be improved.
Manuscript was edited with native experts.
Line 17 Remove (Vit-uh-lie-go)
Yes, we have removed it.
The objective must be clear… Analyze the SNPs (A-89T, C389T and C419T) of the CAT gene in subjects with vitiligo from the Saudi population. Line 21-22
The objective was rewritten (14-16) in the abstract.
The results must be clear:
Line 28-29 page 1:
Haplotype frequency estimation revealed a significant association (p=0.003) and no association with haplotype association with response with any of the alleles. It is not clear what the authors are trying to say.
We have updated the results section to make it easier to understand.
Line 30-32 In the conclusion, the description of the polymorphisms must be clear… A-89T, C389T It is recommended to use the “rs” to identify each polymorphism.
We have updated the sentences based on your suggestions.
Introduction
Page 1 line 44… Vitiligo is categorized into segmental and non-segmental vitiligo. REPLACE BY non-segmental forms.
We have modified the sentence.
Page 1 line 46: (Bin-Saif et 46 to 2021) put correct citation format in the text.
Citation was updated.
Page 2 line 55 says CATALASE and should say Catalase. Remember to correctly place protein and gene names throughout the text.
We have updated the word CATALASE to Catalase. The protein was denoted by CAT and gene was denoted as CAT.
Page 2 lines 68 and 69 include the rs of the 3 SNPs to analyze.
We have added the rs numbers for all the three SNPs.
Matodologia: The writing must be improved
Manuscript was edited with native experts.
Page 2 line 75-76: The participants who were willing to give the oral consent form were excluded from this study. This line can be removed or is only included as exclusion criteria.
We have removed this sentence as per your recommendation.
The participant section must be clearly written. The part of signing the informed consent and ethics committee approval can be included at the end of this section.
We have followed your feedback and updated in the revised manuscript.
Page 2-line 83 pass (Bin-Saif et al 2021) to correct reference format.
We have updated in the revised manuscript.
Add after this clearly the inclusion criteria.
A reference was given for our previously published articles and edited the manuscript with native experts.
Page 2 line 90 and 91: In this study, 2 ml of EDTA blood was collected after the signature obtained from the patient on the informed consent form. Remove, and only leave the content in nucleic acid analysis.
We have updated the sentence in the revised manuscript.
Page 2 line 95 and 96 the citation of the real Qiagen protocol should be added and the citation of (Bin-Saif et al 2021) removed.
We will add the Qiagen protocol leaflet in the supplementary materials
Page 2 line 98 to 100, page 3 line 101
The SNPs in the CAT gene are responsible for several oxidative stress and reactive oxygen species and based on previous studies or bioinformatic analysis, three SNPs such as A89T, 389T and C419T were investigated in vitiligo subjects. It does not provide additional information. Just indicate that the 3 SNPS are going to be analyzed and add their respective “Rs”.
We have added the rsnumbers as per your suggestion.
Genetic analysis:
Page 3 lines 102 to 103:
The PCR protocol must be clear, it must indicate the number of PCR cycles, alignment time of the primers, describe the origin of all the reagents (Primers, restriction enzymes), the brand and model of the equipment used (Thermocycler, Photo-Doc, etc).
All of the suggested information has been updated.
You must replace “run” with “analyzed”
We have updated the error.
Page 3 lines 115 to 119. In the validation you must indicate the type of sequencing equipment used, and preferably include an example image of some electropherogram obtained with the polymorphism analyzed. Furthermore, the reference must correspond exactly to the method; you point out that it is Based on Bogari et al (reference 9), but in that article other authors are cited: “Validation was performed with Sanger sequencing analysis (Fig. 2) based on our recent publication (Alharbi et al., 2021)”.
We have deleted this paragraph.
Page 3 lines 120 to 130. Statistical analysis. The writing needs to be improved. The statistical tests used must be clearly explained: for continuous and discrete variables (continuous variables, indicate whether normality tests were performed before deciding to use the Student's T test). The reference used to support the HW equilibrium calculation is spurious.
We have modified the required sentence and this manuscript was edited with native experts
Include support reference for SNPstat, and I suggest reviewing a similar article on the subject.
We have used SNPSTAT in our previous work (Bin Saif et al 2021)
Results
The interpretation of the results should be improved.
Manuscript was edited with native experts
Page 3 lines 132 to 139 and table 2. The interpretation of the lack of difference in the ages of the vitiligo and controls is not an association result, it should only be used to support that the following analyzes are correct, since there are no differences in the study group.
Yes, we agree with you.
Table3. It is necessary to review the interpretation of the P-value in HW equilibrium, since the P-values for two polymorphisms in Vitiligo cases are not in equilibrium.
Table 3 shows the P values for HWE analysis for all three SNPs in both cases and controls. In addition, we have included HWE and genotyping analysis for CAT gene SNPs in section 3.2.
On the other hand, for the polymorphism analysis it would be important to consider including the chi2 values in the comparison made for 2 groups in each of the models and in the comparison of the alleles.
Chi-Square values were added in the table for all the 3 SNPs
In addition, it is recommended to review the approximation of the values; For example, the authors report for rs7943316 AA vs AT a value of p= 0.01, and it really is p=0.01539, that is, it is close to p=0.02.
We have opted the first 2 digits after the decimal value (as per the recommendation by the statistician)
In alleles of the first 2 polymorphisms, the % symbol is missing.
We have added in the revised manuscript.
Page 5, lines 153 to 186 it is not necessary to repeat the results of the table. It is recommended to limit to describing only relevant data. In this regard, the description is confusing and has errors in interpretation. For example: Do not repeat genotype and allele frequency results for each polymorphism.
Reviewer 1 has suggested some modifications and we have updated it.
Lines 158 to 160 says: and homozygous TT (OR-1.31; 95%CI: 0.61-2.77; p=0.49) and other genetic models (AA+AT vs TT; OR-0.52; 95%CI: 0.33-0.83; p=0.006 and AA+AT vs TT; OR-1.46; 95% CI: 159 0.85-2.48; p=0.16) showed the negative association. And should say for the genetic model: “models (AA+TT vs AT; OR-0.52; 95%CI: 0.33-0.83; p=0.006 and AA+AT vs TT OR-1.46; 95%CI: 159 0.85-2.48; p=0.16), where in the first if there is association”.
We have added your suggestion
Page 5 line 170 says p=01.8 correct
We have updated the error
Line 17º1 to 175 it is not necessary to describe information that is already in the table. On the other hand, the authors should explain their interpretation of risk and protective allele.
We have deleted the suggested sentences
For rs11032709 there are only 2 genotypes. We suggest removing the TT results and the analyzes of genetic models that include them from Table 3. They do not provide information and cannot suggest models. For this reason, lines 176 to 186 must be corrected and redrafted.
Reviewer 1 has suggested a comment and this manuscript was edited with native experts.
In Linkage Disequilibrium and Haplotype analysis it is suggested to explain what they mean by their results, if this affects their analysis. Once again, the text is only a repetition of the result present in tables 4 to 7, and does not provide additional information to understand the objective of its analysis.
Linkage disequilibrium and Haplotype analysis was updated in the revised manuscript. Based on the obtained data available in tables, we have explained in the results section.
Figure 1. The additional information for each of the samples must be in text format as a caption, not as part of the photograph. It is recommended to describe characteristics of the molecular weight marker, and its representative sizes must be described in the image.
We have modified the figure as per your suggestion
For Figure 2, it must contain a sufficiently descriptive image caption, in addition, the image format, separated for one analysis and the other, does not provide relevant information. They should indicate Figures 2a and 2b, each with an explanation of its content.
We have updated the figure-2 as per your recommendations
Discussion:
The discussion should contain the contrast of the results obtained in this work with what is already published. In this regard lines 200 to 215 can be deleted. Lines 215 to 230 is a repetition of the results.
We have updated the discussion as per your suggestion
Only from line 231 is there a real discussion of the work.
We have included
Line 238 still needs to be added reference to He et al meta-analysis studies.
Reference was updated in the revised manuscript.
Line 251 to 253 actually describe the result of the study carried out in the Korean population and how it compares with what was obtained in this investigation.
Our study SNPs was previously studied in Korean population and I am explaining their study in detail. Our study results were already discussed in the above paragraphs of the discussion
Lines 254 to 259 delete. Just leave a section of study limitations.
We have deleted the sentence as per your suggestion.
Dear editor, the line numbers were shuffled based on adding and deleting the sentences. If we miss any of the modifications, it was not our intension but skipped them accidentally due to the confusion in the line numbers. We apologize for the delay.
The conclusion should be improved
We have modified the grammatical errors
If the authors consider the reference genotypes, an association is observed in each polymorphism. On the other hand, they could mention that considering certain genetic models, there is also an association between genotypes and the presence of vitiligo.
We have recommended the heterozygous genotype and dominant model in A89T and C389T SNPs in CAT gene.
Lines 268 to 270 should be removed as it is not a conclusion to your work.
We have deleted the suggested sentence.
The bibliography should be reviewed. There are lost references in the text, and others that do not correspond (example reference 10)
The references were updated.
Our Team is thankful to “Prince Naif Health Research Centre, Investigator Support Unit for providing the Language editing services”
Round 2
Reviewer 2 Report
The article previously entitled “Evaluation of Catalase genetic variants associated with Vitiligo in Saudi subjects” Presents the result of the association analysis of three SNPs in the Catalase gene and the presence of Vitiligo in this population.
This paper contains a revised version of the original paper, however it still maintains some errors in the interpretation of the results, both text and additional references were added, some of which did not provide relevant information, so they need to be addressed.
In the abstract.
Lines 24 to 27 of results. It is necessary to improve the wording; something like:
The rs7943316 and rs11032709 SNPs of CAT gene showed a positive association with vitiligo for both Heterozygous genotypes and dominant genetic models…
Lines 29 and 30
It should say Conclusions: The rs7943316 and rs11032709 SNPs of CAT gene were strongly associated with the susceptibility to vitiligo
Introduction
Page 3 lines 120 and 121 review wording. It is suggested:
Few studies have investigated the association between vitiligo susceptibility and SNPs in the CAT gene, and in the Saudi population it has not been investigated so far.
Methodology:
Study participants. Page 3 Lines 133 to 135. They must explain how the sample size was calculated. I think there is an error in the interpretation. If you calculate 311 participants, these are the cases... and at least the same number of controls. Therefore, I suggest that you describe the formula for calculating the sample size used with the parameters considered in this.
Page 4 lines 156 to 184.
Authors must include company data and country of origin for each of the reagents, kits and equipment used. Again missing for Qiagen, NanoDrop, They do not indicate the Thermocycler Brand for PCR; and it is not indicated with what type of equipment the result was photodocumented (for example, type of UV transilluminator)
Page 5 Figure 1. It is recommended that you improve your image, it presents missing parts, its quality is worse than the previous ones and it still preserves data in the photograph of the content of each sample.
Page 5 lines 211 to 218. It would be convenient if they explained the reason for omitting the validation
Statistic analysis. Page 5 lines 219 to 221. The author is insisted that they have to justify the use of a statistical test: when performing an analysis of means it is imperative to perform a normality test, with which it is defined whether a Student's T test or Mann-Whitney U. In addition to the age data presented, the interpretation presented by the authors in the results is incorrect. In this regard, the analysis of the mean and standard deviation allow us to define that both groups do not have a significant difference, therefore, the remaining analyzes do they can drive in these groups, since age would not be affecting the validity of their results. For this reason, it is not correct to point out that age has nothing to do with vitiligo. It is suggested that they request statistical support in the interpretation of their results. (Page 6 lines 233-236)
Table 3: Already in the previous review, comments were made about the presentation of the data, but its description is still confusing.
First, when you test 3 genotypes by chi-square test, the first test indicates the test results for all three genotypes. In this case, it is a 3x2 test, so it only provides chi square and P value. Authors should include that first value. Then they can make all the comparisons that they already present in the table against the reference genotype, which you present as 2x2 tests where you can add CI and OR.
Second. I understand that whoever helped you in the analyzes suggested you include only two decimal places, however you must understand that you are making approximations incorrectly. For example, the authors report for rs7943316 AA vs AT a value of p= 0.01, and it really is p=0.01539, that is, it is close to p=0.02. Also, if your answer is still occupying two decimal places, why for the polymorphism rs769217 and rs11032709 consider P values with 3 and up to 4 decimal places?
I recommend correcting the table.
Page 7, lines 258 to 271, page 8 lines 272 to 282. A correct interpretation of the data is missing. If you present a genotype or genetic model that presents a positive or negative association, your interpretation should be associated or not associated with vitiligo. I think it is necessary that the statistical analysis be focused on that. In other words, they must explain what genotype or genotype sum is associated with vitiligo. Statistical help is required, not just a native expert reviewing the text.
Page 8 line 295 to page 9 line 322.
Linkage disequilibrium analysis.
In genetics, the property of some genes of genetic populations of not segregating independently is called linkage disequilibrium or LD, that is, they have a recombination frequency of less than 50%. In this regard, between different polymorphisms located on the same chromosome and relatively close to each other, a certain degree of correlation or statistical association called linkage disequilibrium is usually observed. Therefore, your interpretation is not correct. This is not to say that SNPs have a role in vitiligo or not, but rather that the alleles segregate together in subjects who do or do not have vitiligo.
Page 10 lines 324-328.
An analysis of the haplotype frequency estimation revealed a significant association (p=0.003) and no association with haplotype association in response to any 326 of the alleles (Tables 6 and 7).
The wording is not understandable. What do they want to say?
Discussion.
It must be improved. Although the authors attended to previous comments and suggestions, the wording is not correct, they omit words, and the sentences do not make sense. It is suggested that an already corrected document be included in order to review the wording.
Line 333 to 357 does not contribute to the discussion, it is only an introduction that it is CAT. The discussion should really start from line 358, since in this section the comparison between the result of this work and previous studies should be presented.
Lines 388 to 400 They must be eliminated, this does not contribute anything to the discussion. It would only be interesting if you intend to make a historical description of the disease, but it does not make sense if what you are comparing is the genetic factor in CAT.
On the other hand, when they compare with meta-analyses, it is enough that they indicate whether their results are similar or not, it is not necessary that they include all the values obtained in that study.
Conclusion:
I insist that the authors request statistical advice.
Regarding the paragraph: Because this study included elderly older controls, the findings should be interpreted with caution. These types of sentences demonstrate ignorance of the data.
When you compare the groups and observe that the P is greater than 0.05, it indicates that the groups are similar (As long as you ensure that a statistical test of normality was applied before deciding if it was correct to use the T test. Therefore, if Subsequent analyzes can be conducted if age is not a bias factor.
With the phrase that you indicate, you will be indicating that your study could not be carried out, and that its results are incorrect. I recommend that you reevaluate your interpretation and seek professional statistical help.
additional tip
Regarding a Data Availability Statement: All the obtained data is present in the manuscript.
The availability of data refers to the fact that if someone asked you for the information, are you willing to provide it?, that is, if you are willing to provide the database that you used to carry out the analyses, not just the interpretation that you publish.
Author Response
The article previously entitled “Evaluation of Catalase genetic variants associated with Vitiligo in Saudi subjects” Presents the result of the association analysis of three SNPs in the Catalase gene and the presence of Vitiligo in this population. This paper contains a revised version of the original paper, however it still maintains some errors in the interpretation of the results, both text and additional references were added, some of which did not provide relevant information, so they need to be addressed.
- A) Dear Reviewer,
We all the authors as a team, we are grateful for your assistance in thoroughly screening our revised manuscript. We worked on it in response to your comments, and we believe we have addressed all of your concerns. Thank you for your continued interest in reviewing our manuscript.
In the abstract.
Lines 24 to 27 of results. It is necessary to improve the wording; something like: The rs7943316 and rs11032709 SNPs of CAT gene showed a positive association with vitiligo for both Heterozygous genotypes and dominant genetic models…
- A) We modified the sentence and added the recommended sentence based on your suggestion.
Lines 29 and 30
It should say Conclusions: The rs7943316 and rs11032709 SNPs of CAT gene were strongly associated with the susceptibility to vitiligo.
- A) We have also included your suggested sentence here.
Introduction
Page 3 lines 120 and 121 review wording. It is suggested:
Few studies have investigated the association between vitiligo susceptibility and SNPs in the CAT gene, and in the Saudi population it has not been investigated so far.
- A) Once again, we have incorporated your suggested sentence.
Methodology:
Study participants. Page 3 Lines 133 to 135. They must explain how the sample size was calculated. I think there is an error in the interpretation. If you calculate 311 participants, these are the cases... and at least the same number of controls. Therefore, I suggest that you describe the formula for calculating the sample size used with the parameters considered in this.
- A) In this study, 152 Vitiligo cases and 159 control subjects were recruited, yielding a total of 311 (152+159 = 311) participants when the two numbers are added together. Our intension was to accumulate 200 Vitiligo cases and 200 controls. This study was conducted during pandemic criteria (2020-2021), and patient visits to the clinic were limited, so we were unable to collect our target sample size, but we did our best to collect the samples. We have already published a manuscript based on 152 cases of vitiligo and 159 controls, which you can find here
(https://pubmed.ncbi.nlm.nih.gov/35380061/).
Page 4 lines 156 to 184.
Authors must include company data and country of origin for each of the reagents, kits and equipment used. Again, missing for Qiagen, NanoDrop, they do not indicate the Thermocycler Brand for PCR; and it is not indicated with what type of equipment the result was photo documented (for example, type of UV transilluminator)
- A) We apologies for missing the details in the main document.
In this study, DNA was isolated using Qiagen (Cat#51104, Lot#157037757; 40724 Hilden, Germany). NanoDrop was performed a spectrophotometer (Thermoscientific, NANODROP ONEC, S. No-AZY1707157; Thermo Fischer Scientific, Madison, USA). PCR was performed with 96-well thermal cycler (Applied Biosystems, Model # 9902, Serial # 2990210130, Singapore). PCR reaction was carried out using 50ul reaction with Qiagen Taq PCR master mix (CAT# 201445; 40724 Hilden, Germany). PCR products were separated on a 2% agarose gel (Lonza, SeaKem® LE Agarose, CAT # 50004, Rockland, USA). PCR products were digested with specific restriction enzymes from NEB (New England Biolabs, Ipswich, UK).
As per our knowledge, all the details were documented in the revised manuscript and highlighted with yellow color.
Page 5 Figure 1. It is recommended that you improve your image, it presents missing parts, its quality is worse than the previous ones and it still preserves data in the photograph of the content of each sample.
- A) The previous image has been saved. If you do not converse, we will remove the figure-1.
Page 5 lines 211 to 218. It would be convenient if they explained the reason for omitting the validation
- A) We have an ABI Prism 310 genetic analyzer in our lab. Because this instrument was obsolete after 2020, Integrated Gulf Biosystem (Riyadh, Saudi Arabia) has retained the services.
Statistical analysis. Page 5 lines 219 to 221. The author is insisted that they have to justify the use of a statistical test: when performing an analysis of means it is imperative to perform a normality test, with which it is defined whether a Student's T test or Mann-Whitney U. In addition to the age data presented, the interpretation presented by the authors in the results is incorrect. In this regard, the analysis of the mean and standard deviation allow us to define that both groups do not have a significant difference, therefore, the remaining analyzes do they can drive in these groups, since age would not be affecting the validity of their results. For this reason, it is not correct to point out that age has nothing to do with vitiligo. It is suggested that they request statistical support in the interpretation of their results. (Page 6 lines 233-236)
- A) When compared to the cases, the controls were older. We exclude student t-test details from the statistical analysis (pages 220-221) and p values from the results (page no 238)
Table 3: Already in the previous review, comments were made about the presentation of the data, but its description is still confusing.
- A) We have discussed the P values for HWE analysis in the results section 3.2 based on previous comments. The results section is structured into four paragraphs. The first paragraphs discussed HWE analysis/results, the second paragraph discussed A-89T (rs7943316) results, the third paragraph discussed C389T (rs769217) results, and the final paragraph discussed C419T (rs11032709) results. If you have any specific recommendation, please let us know so that we can amend as per your suggestion.
First, when you test 3 genotypes by chi-square test, the first test indicates the test results for all three genotypes. In this case, it is a 3x2 test, so it only provides chi square and P value. Authors should include that first value. Then they can make all the comparisons that they already present in the table against the reference genotype, which you present as 2x2 tests where you can add CI and OR.
- A) We have now included chi-square values in all three SNPs described in section 3.2 results for HWE analysis and genotype/allele frequencies.
Second. I understand that whoever helped you in the analyzes suggested you include only two decimal places, however you must understand that you are making approximations incorrectly. For example, the authors report for rs7943316 AA vs AT a value of p= 0.01, and it really is p=0.01539, that is, it is close to p=0.02. Also, if your answer is still occupying two decimal places, why for the polymorphism rs769217 and rs11032709 consider P values with 3 and up to 4 decimal places?
- A) Based on your recommendations, we have increased the decimal value of 4 in the p values and updated table 3.
I recommend correcting the table.
- A) We have also updated in the table 3.
Page 7, lines 258 to 271, page 8 lines 272 to 282. A correct interpretation of the data is missing. If you present a genotype or genetic model that presents a positive or negative association, your interpretation should be associated or not associated with vitiligo. I think it is necessary that the statistical analysis be focused on that. In other words, they must explain what genotype or genotype sum is associated with vitiligo. Statistical help is required, not just a native expert reviewing the text.
- A) We explained the genotype and allele frequencies in both vitiligo cases and controls for the A-89T (rs7943316) and C389T (rs769217) SNPs, as well as the genotype, genetic models (i.e., dominant, co-dominant, and recessive models), and allele frequencies compared between vitiligo cases and controls. However, in response to your feedback, we revised the manuscript. If you have any suggestions, we can incorporate in the revised manuscript.
In the results
Page 8 line 295 to page 9 line 322.
Linkage disequilibrium analysis.
In genetics, the property of some genes of genetic populations of not segregating independently is called linkage disequilibrium or LD, that is, they have a recombination frequency of less than 50%. In this regard, between different polymorphisms located on the same chromosome and relatively close to each other, a certain degree of correlation or statistical association called linkage disequilibrium is usually observed. Therefore, your interpretation is not correct. This is not to say that SNPs have a role in vitiligo or not, but rather that the alleles segregate together in subjects who do or do not have vitiligo.
- A) We have now rectified that error. Thank you for your comment.
Page 10 lines 324-328.
An analysis of the haplotype frequency estimation revealed a significant association (p=0.003) and no association with haplotype association in response to any 326 of the alleles (Tables 6 and 7). The wording is not understandable. What do they want to say?
- A) In Table 6, the analysis of haplotype frequency estimation revealed a significant association (p=0.003). However, there was a negative association with haplotype association in response to any of the alleles in Table 7.
Discussion.
It must be improved. Although the authors attended to previous comments and suggestions, the wording is not correct, they omit words, and the sentences do not make sense. It is suggested that an already corrected document be included in order to review the wording.
- A) Now, we have edited and improved the discussion.
Line 333 to 357 does not contribute to the discussion; it is only an introduction that it is CAT. The discussion should really start from line 358, since in this section the comparison between the result of this work and previous studies should be presented.
- A) We have deleted the sentences from 333-357 as per your suggestion.
Lines 388 to 400 They must be eliminated, this does not contribute anything to the discussion. It would only be interesting if you intend to make a historical description of the disease, but it does not make sense if what you are comparing is the genetic factor in CAT.
- A) We removed the sentences from 388-400 based on your suggestion.
On the other hand, when they compare with meta-analyses, it is enough that they indicate whether their results are similar or not, it is not necessary that they include all the values obtained in that study.
- A) We have removed their values
Conclusion:
I insist that the authors request statistical advice.
- A) We have added in the conclusion.
Regarding the paragraph: Because this study included elderly older controls, the findings should be interpreted with caution. These types of sentences demonstrate ignorance of the data.
- A) This sentence has been removed.
When you compare the groups and observe that the P is greater than 0.05, it indicates that the groups are similar (As long as you ensure that a statistical test of normality was applied before deciding if it was correct to use the T test. Therefore, if Subsequent analyzes can be conducted if age is not a bias factor.
- A) We agree with you, and the specific sentence has been removed from the conclusion.
With the phrase that you indicate, you will be indicating that your study could not be carried out, and that its results are incorrect. I recommend that you reevaluate your interpretation and seek professional statistical help.
- A) To avoid misunderstanding, we have removed the final sentence and recommendation from the conclusion.
additional tip
Regarding a Data Availability Statement: All the obtained data is present in the manuscript.
The availability of data refers to the fact that if someone asked you for the information, are you willing to provide it?, that is, if you are willing to provide the database that you used to carry out the analyses, not just the interpretation that you publish.
- A) Data set will be provided based upon the request. Thank you for your comment. We have updated in the revised manuscript.
